# Changes in Essential Oil Composition, Polyphenolic Compounds and Antioxidant Capacity of Ajowan (*Trachyspermum ammi* L.) Populations in Response to Water Deficit

**DOI:** 10.3390/foods11193084

**Published:** 2022-10-05

**Authors:** Gita Mirniyam, Mehdi Rahimmalek, Ahmad Arzani, Adam Matkowski, Shima Gharibi, Antoni Szumny

**Affiliations:** 1Department of Agronomy and Plant Breeding, College of Agriculture, Isfahan University of Technology, Isfahan 84156 83111, Iran; 2Department of Horticulture, College of Agriculture, Isfahan University of Technology, Isfahan 84156 83111, Iran; 3Department of Pharmaceutical Biology and Biotechnology, Wroclaw Medical University, 50-556 Wroclaw, Poland; 4Core Research Facilities (CRF), Isfahan University of Medical Sciences, Isfahan 81746 73461, Iran; 5Department of Food Chemistry and Biocatalysis, Wrocław University of Environmental and Life Sciences, 50-375 Wroclaw, Poland

**Keywords:** ajowan, drought stress, essential oil, thymol, polyphenols components

## Abstract

Ajowan (*Trachyspermum ammi* L.) is considered a valuable spice plant with a high thymol content. Seed yield, essential oil constituents, polyphenolic composition, and antioxidant capacity of ajowan (*Trachyspermum ammi* L.) populations were evaluated in three (normal, moderate, and severe) water irrigation regimes. The highest essential oil content (5.55%) was obtained under normal condition in the Yazd population. However, both essential oil and seed yield showed significant reductions as a result of water stress. According to gas chromatography–mass spectrometry (GC–MS) analysis, thymol (61.44%), γ-terpinene (26.96%), and *p*-cymene (20.32%) were identified as the major components of the oil. The highest (89.01%) and the lowest (37.54%) thymol contents were in Farsmar and Hamadan populations in severe stress condition, respectively. Based on HPLC analysis, chlorogenic (3.75–47.35 mg/100 g), caffeic (13.2–40.10 mg/100 g), and ferulic acid (11.25–40.10 mg/100 g) were identified as the major phenolic acids, while rutin was determined as the major flavonoid (11.741–20.123 mg/100 g). Moreover, total phenolic and flavonoid contents were elevated under drought stress treatment, while antioxidants responded inconsistently to stress based on two model systems. Overall, the Yazd population exhibited a superior response to water stress, as evidenced by its less reduced thymol and oil yield content, while Arak and Khormo had the highest accumulation of polyphenolic compounds.

## 1. Introduction

Thymol is the major component in many of medicinal and aromatic plant leaves, such as thyme and oregano [1,2], but in ajowan, it accumulates in the seeds [3], which might be beneficial for certain types of food processing. Moreover, high flavonoid and phenolic contents of ajowan seeds can provide new insights for using it as nutraceutical food as well as pharmaceutical products [4]. Finally, by improving the content of essential oils, thymol, flavonoids, and phenolics can be beneficial for health-related applications [5,6].

Ajowan (*Trachyspermum ammi* L.), a member of the Apiaceae (Umbelliferae) family, originated from Eastern Mediterranean areas and was later spread to India and other countries in the course of the Greek conquest of Central Asia [7]. Commonly used as an important seed spice [8,9], the plant is also used as a dietary supplement due to its health-related properties such as antioxidant, antimicrobial, sedative, carminative, antifungal, antiplatelet–aggregatory, and antispasmodic effects [10,11,12].

The content and composition of secondary metabolites of plants can be highly influenced by such environmental factors as growth (soil, climate, and management), harvesting (phenological stage), and storage conditions [13,14]. Abiotic stresses are also important environmental factors that might lead to enhanced production of secondary metabolites in the species as a means of increasing the plant’s resilience [13]. In addition to the essential oil compounds, the seeds of ajowan contain polyphenolic components that belong to one of the most widely used groups of phytochemicals with multiple physiological activities [15]. Many phenolic compounds possess high antioxidant properties due to their ability to trap free radicals by forming stable phenoxylates. The main role of antioxidants is neutralizing free radicals, which are active and harmful substances for humans [16].

As a major adverse environmental stressor, drought forms an important concern for overall agricultural production in arid and semi-arid regions, which might even increase in the future. The effects of drought stress on essential oil constituents and polyphenolic components of plants have already been reported in many medicinal plants of Apiaceae, including *Foeniculum vulgare* [17], *Coriandrum sativum* L. [18], *Crithmum maritimum* L. [19], *Eryngium* species [20], and *Cuminum cyminum* L. [21]. Most previous studies of ajowan in Iran highlighted the variations in essential oil components [9,22]. Moreover, the effects of water stress on the polyphenolic content and phytochemical composition of ajowan populations have not been studied.

Thus, the aims of the current study were (1) to explore the effects of genotype and water stress on the essential oil content and composition, (2) to evaluate polyphenolic accumulation in response to water stress in different ajowan populations, and (3) to use multivariate analyses to indicate potentially superior populations. The analyses were based on GC–MS and HPLC chromatographic techniques.

## 2. Materials and Methods

### 2.1. Plant Materials

Twelve Iranian ajowan populations were cultivated under three irrigation regimes. The seeds were obtained from the Gene Bank of the Research Institute of Forests, Range, and Watershed Management Organization, and the populations were identified by Dr. Valiolah Mozaffarian from this institute using the *Flora Iranica* [23].

### 2.2. Irrigation Regimes

The seeds were sown in a randomized complete block design using three irrigation regimes replicated three times in an experimental field at the Lavark Research Farm of Isfahan University of Technology. The soil texture was a clay loam with a bulk density of 1.4 g cm^−3^ and a pH of 7.8. The plot size was 1 × 2 m^2^, and the individual plants were spaced 30 cm apart. Managed allowable depletion (MAD) of the soil available water (SAW) was used to determine the irrigation treatments. Soil water content was evaluated using the soil moisture curve based on TDR (time domain reflectometry device), and the soil water content was determined according to the method described in [24]. These provisions yielded irrigation regimes based on 40% (non-stress), 60% (moderate water stress), and 80% (severe water stress) MAD of SAW in ajowan populations. Finally, the treatments were initiated on 5 June 2017, that is, at the beginning of the flowering stage, and continued throughout the seed filling stage until 10 July 2017.

### 2.3. Essential Oil Distillation

Powdered seed (approximately 50 g) material was used for hydro-distillation for 6 h using a Clevenger-type apparatus. EO yield was evaluated according to the following formula [25]:(1)$$\mathrm{EO}\;\mathrm{yield}\;(\%)=\frac{\mathrm{volume}\;\mathrm{of}\;\mathrm{EO}\;\mathrm{obtained}\;\left(\mathrm{mL}\right)}{\mathrm{mass}\;\mathrm{of}\;\mathrm{dry}\;\mathrm{matter}\;\left(g\right)}\;\times \;100$$

### 2.4. GC–MS Analysis

The composition of the volatile constituents of the essential oil was analysed using an Agilent 7890 gas chromatograph (Agilent Technologies, Palo Alto, CA, USA) with an HP-5MS 5% phenylmethylsiloxane capillary column (30 m × 0.25 mm, and a film thickness of 0.25 μm). The analyses were accomplished using helium as the carrier gas in a split ratio of 1:20 at a flow rate of 2 mL min^−1^. The initial oven temperature was optimized at 60 °C for 3 min and ramped at 3 °C min^−1^ to 120 °C before it was raised to 300 °C at 15 °C min^−1^. The injector temperature was maintained at 300 °C. An Agilent 5975 C (Agilent Technologies, Palo Alto, CA, USA) mass detector was applied. The scanning conditions comprised 39–400 m/z, 200 °C, and an electron ionization of 70 eV. Injection volume was set at 1 μL of 0.1% solution of EOs in cyclohexane.

#### Identification of Essential Oil Constituents

EO constituents were determined according to comparison: (a) mass spectrum of unknown compounds with spectra presented in NIST 17 (National Institute of Standards and Technology), Wiley 275. L, and literature data [26]; (b) logarithmic retention indices (RI) in relation to a *n*-alkanes series (C8–C24) with data published in the NIST17 database and Adams, R.P. (2007), *Identification of essential oil components by gas chromatography/mass spectrometry* (Vol. 456, pp. 544–545). Carol Stream: Allured publishing corporation; (c) retention times of available standards. The minimum match value for MS search was set to 90%. The percentage of identified compounds in EOs was based on GC–MS chromatograms peaks.

### 2.5. Methanolic Extract and Total Phenolic Content

Total phenolic content (TPC) was measured according to the method described by Gharibi et al. (2016) [27]. For this, eight grams of the dried material was extracted by 200 mL of methanol (80%) using a shaker run at 150 rpm for 24 h at 25 °C. Then the extracts were filtered, and the steps were repeated three times. The reaction mixture included 2.5 mL of Folin–Ciocalteu reagent, 0.5 mL of extract, and 2 mL of sodium carbonate (7.5%). Finally, the absorbance was measured at 765 nm, and the amount of TPC was reported as tannic acid equivalent per gram dry weight.

### 2.6. Total Flavonoid Content (TFC)

The aluminum chloride colorimetric method was applied for the determination of TFC [28]. First, 75 μL of NaNO_2_ solution (5%) was mixed with 125 μL of the extract. The blend was set for six minutes. Then, 150 μL of AlCl_3_ (10%) was added and incubated for 5 min, and finally 750 μL of NaOH (1 M) was added. The absorbance of pink extract was evaluated at 510 nm. TFC was presented in mg of quercetin equivalents (QEs) per gram of the extract.

### 2.7. Antioxidant Activity

#### 2.7.1. DPPH Assay

A DPPH assay was performed using the method reported by Tohidi et al. (2017) [25]. For this, plant extracts (0.1 mL) were mixed at different concentrations of 50, 100, 300, and 500 ppm with 5 mL of 0.1 mM methanol DPPH solution, and absorbances were measured at 517 nm. BHT was also used as the synthetic antioxidant. Finally, antioxidant capacity was reported as EC50 value.

#### 2.7.2. Reducing Power

The ability to reduce iron ions was assessed based on the method described in [25]. Accordingly, 2.5 mL of the extracts with (50, 100, 300, and 500 ppm) concentrations were added to 2.5 mL of 0.2 M phosphate buffer and 2.5 mL of 1% potassium ferricyanide (K_3_Fe (CN)_6_) solution. After 20 min at 50 °C, 2.5 mL of 10% trichloroacetic acid was added to the mixture and centrifuged at 3000 rpm for 10 min. The supernatant (2.5 mL) was mixed with 2.5 mL of distilled water and 0.5 mL of 0.1% iron (III) chloride. Finally, absorbance was read at 517 nm using BHT as the synthetic antioxidant.

### 2.8. HPLC Analysis

The extracts were prepared for HPLC (model Agilent 1090) analysis, based on Gharibi et al. [29]. Accordingly, 2.5 g of ajowan well-powdered seeds was extracted with 50 mL methanol (HPLC grade, Merck). HPLC was done based on the procedure described in [29]; briefly, the extracts were dissolved in 80% (*v*/*v*) methanol. Calibration curves of all polyphenolic standards, viz., chlorogenic acid, ferulic acid, caffeic acid, rosmarinic acid, *p*-coumaric acid, gallic acid, rutin, and apigenin, were prepared in concentrations of 0.2–20 mg/L (Appendix A). The chromatograms were acquired at 268–370 nm. Twenty microliters of the extract were injected into the analytical column (250 mm × 4.6 mm, 5 μm, Waters Symmetry C18 column with fitting 10 mm × 4 mm precolumn, Waters Corp., Milford, MA, USA). The gradient mobile phase was composed of solvent A (0.1% aqueous formic acid) and B (0.1% formic acid) in acetonitrile, with the detector wavelength set in the range of 200–400 nm. The gradient conditions at a flow rate of 0.8 mL min^−1^ were as follows: a linear step from 10% to 26% solvent B (*v*/*v*) in 40 min, 65% solvent B at 70 min, and finally to 100% solvent B at 75 min. The phenolic components of seed extracts were identified by comparison of the UV spectra and retention time (RT) with those of standards. Quantitation was based on the peak areas using calibration curves for each used standard. The results were reported as mg per 100 g of sample dry weight.

### 2.9. Statistical Analysis

All the tests were performed in triplicate. The data obtained were expressed as means ± standard deviation (SD). Statgraphics Software (ver. 18) was used for graph plotting. and clustering analysis (HCA) was performed using SAS JMP ver. 11.

## 3. Results

### 3.1. Seed Yield

The highest seed yield was obtained in normal conditions, which subsequently decreased significantly with declining water availability. The loss of seed yield under drought stress was attributed to the adverse effects of drought stress on seed yield components. The highest percentage of seed yield was obtained under moderate drought stress (5059.20 g/m^2^) in Esfahfo populations. The seed yield ranged from 714.38 g/m^2^ in Arakkho populations to 4841.55 g/m^2^ in Yazdshah populations under normal conditions. Seed yield varied from 274.32 g/m^2^ in Farsfars populations under moderate stress treatment to 5059.20 g/m^2^ in Esfahfo populations under the same conditions. Similar trends have been reported for other medicinal plants, including *Nigella sativa* [30], *Mentha piperita*, *Lavandula latifolia*, *Thymus mastichina*, *T. capitatus*, *Salvia sclarea*, and *S. lavandulifolia* [31].

A consistent decrease in seed yield was observed with increasing drought, with the highest decline in seed yield (66 g/m^2^) under the severe drought stress in Qazvin populations. A lower photosynthesis rate can result in reduced dry matter. Moreover, the seed development may be affected by the shorter seed filling period as a result of drought treatment during the early maturity stage [32]. Seed filling is highly affected by such metabolic processes as photosynthesis, assimilates translocation as well as availability of precursors for the biosynthesis of seed components. Previous studies highlighted the sensitivity of these processes to environmental stresses that ultimately led to changes in the balance between enzymes and transporters in the vegetative and reproductive organs [33].

### 3.2. Essential Oil Yield

Introduction of high oil-yield cultivars promises enhanced efficiency of essential oil production and improved food products. In the present study, the highest amounts of essential oil were obtained under normal treatments (5.55% and 5.37%), with 1.78% recorded for Farsfars and 5.55% for Yazd populations (Table 1). Similar amounts of oil yield were reported for ajowan populations [8]. Under the moderate stress treatment, Esfahfo and Hamadan populations exhibited oil yields of 0.7% and 3.34%, respectively. Variations in oil content have been reported in other medicinal plant species under drought conditions. but the reports on seed oil content are somewhat controversial. Essential oil in the *Thymus* species under severe drought stress observed increased [34]. It is generally established that during drought, plants produce lower amounts of essential oil than under normal conditions. This was confirmed by the present results indicating that longer durations of water stress led to decreased amounts of essential oil in ajowan seeds. In contrast, some studies reported increased production of essential oil as a result of water stress [35,36].

### 3.3. Essential Oil Composition

The chemical composition of ajowan populations was influenced by different levels of drought stress. Under the normal irrigation regime, thymol (40.21–61.44%), γ-terpinene (19.46–26.96%), and *p*-cymene (15.09–20.32%) constituted the major essential oil (EO) components (Table 2). Drought stress had different effects on these compounds, such that the highest (50.35%) thymol content was recorded for Khormo populations under moderate stress and the lowest (40.88%) was observed in Ardebil populations. Moreover, Arak and Qazvin populations exhibited the highest (24.64%) and lowest (0.55%) *p*-cymene contents, respectively. Finally, Ardebil and Farsmar populations had the highest (31.63%) and lowest (26.04%) values of γ-terpinene, respectively.

### 3.4. Multivariate Analyses

#### Principle Component Analysis (PCA) of Essential Oils

The biplot of the essential oil compounds in control conditions implied that two main components explained 59.42% of the variation (Figure 1a). Component 1 included myrcene, α-thujene, α-terpinene, *p*-cymene, β-thujone, and γ-terpinene with positive values and TPC with a negative value. Component 2 consisted of β-pinene, myrcene, β-thujone, thymol, carvacrol, and essential oil yield. The studied populations were divided into four groups. In the first group, Khormo had component 1 with a negative value, and TPC was the indicator of this group. In the next group, Yazd possessed the highest values. In the third group, Farsfars was not significant in component 1. Other populations were classified in the fourth group.

**Table 2 foods-11-03084-t002:** Volatile compounds (relative quantity in %) of essential oils of studied ajowan populations.

Compounds		α-Thujene	α-Pinene	Sabinene	β-Pinene	Myrcene	α-Terpinene	*p*-Cymene	γ-Terpinene	*cis*-Sabinenehydrate	β-Thujone	Terpinene-4-ol	Pulegone	Thymol	Carvacrol	Total
* RI ^a/b^		927/929	941/937	972/974	979/979	996/991	1017/1017	1025/1024	1057/1060	1068/1070	1110/1114	1181/1177	1246/1237	1290/1291	1301/1299	
Stress	Populations															
Normal	Arak	0.36	0.15	0.18	1.01	0.35	0.38	18.74	26.89	0	0.81	0.21	0	50.16	0.65	99.89
	Arakkho	0.43	0.17	0.33	1.43	0.53	0.34	17.74	26.05	0	0.61	0.17	0.29	50.99	0.58	99.66
	Ardebil	0.44	0.12	0.31	0.41	0.58	0.34	20.14	26.96	0	0.73	0.27	0.36	48.84	0.5	100
	Esfahfo	0.31	0.11	0.21	0.66	0.47	0.19	20.32	23.15	0	0.74	0.29	0.71	52.2	0.63	99.99
	Farsfars	0.2	0	0.17	0.12	0.24	0.28	20.32	21.37	0	0.45	0	0	40.21	0	83.36
	Farsmar	0.39	0.11	0.26	0.13	0.5	0.28	18.61	22.62	0	0.77	0.2	0.38	54.98	0.64	99.87
	Hamdan	0.37	0.11	0.32	0.56	0.57	0.25	19.46	22.11	0.21	0.74	0.26	0.43	53.91	0.7	100
	Khorbir	0.36	0.13	0.23	0.6	0.57	0.44	18.23	26.33	0	0.74	0.41	0	51.4	0.5	99.94
	Khormo	0.18	0.09	0.14	0.9	0.3	0.12	15.09	19.46	0	0.57	0.19	0	61.44	0.81	99.29
	Qazvin	0.42	0.13	0.25	0.43	0.48	0.34	18.87	22.85	0	0.77	0.31	0	54.29	0.66	99.8
	Yazd	0.31	0.1	0.19	0.19	0.41	0.22	16.96	20.32	0	0.47	0.28	0.62	58.92	0.72	99.71
	Yazshah	0.4	0.12	0.3	0.37	0.55	0.19	19.08	23.02	0	0.75	0.31	0.56	53.65	0.61	99.91
Medium	Arak	0.68	0.21	0.35	1.2	0.48	0.54	24.64	26.19	0.17	0.71	0.17	0.1	43.91	0.44	99.79
	Arakkho	0.68	0.24	0.43	1.66	0.71	0.62	21.19	29.53	0.2	0.73	0.1	0.1	42.65	0.71	99.55
	Ardebil	0.51	0.13	0.27	0.53	0.45	0.41	24.36	31.63	0	0.57	0	0	40.88	0.26	100
	Esfahfo	0.32	0.1	0.21	0.69	0.38	0.4	17.97	30.81	0	0.46	0.12	0	48.2	0.35	100
	Farsfars	0.39	0	0.24	0.16	0.36	0.36	21.44	27.77	0	0.51	0	0	48.78	0	100
	Farsmar	0.55	0.14	0.31	0.35	0.59	0.53	21.09	26.04	0.19	0.62	0.13	0.1	48.6	0.49	99.73
	Hamdan	0.57	0.17	0.35	0.94	0.64	0.55	20.31	28.87	0.19	0.6	0.15	0.1	45.62	0.62	99.68
	Khorbir	0.75	0.22	0.42	1.26	0.69	0.61	23.18	27.09	0.26	0.73	0.15	0.07	43.5	0.82	99.75
	Khormo	0.48	0.18	0.31	1.58	0.57	0.53	17.3	26.76	0.24	0.56	0.15	0.15	50.35	0.53	99.69
	Qazvin	0.5	0.14	0.35	0.64	0.55	0.54	0.55	28.13	0.23	0.58	0.09	0	46.00	1.07	79.37
	Yazd	0.53	0.13	0.34	0.44	0.64	0.61	19.21	28.49	0.22	0.62	0.17	0.14	47.07	0.9	99.51
	Yazshah	0.56	0.15	0.3	0.72	0.55	0.52	21.22	29.98	0.21	0.57	0.16	0	45.06	0	100
Severe	Arak	0.63	0.17	0.42	0.53	0	0.67	20.25	29.74	0.19	0.67	0.23	0.12	44.49	0.65	98.76
	Arakkho	0.57	0.22	0.26	1.81	0.54	0.74	24.98	25.65	0.09	1.29	0.17	0	41.05	0.91	98.28
	Ardebil	0.54	0.13	0.29	0.45	0.56	0.52	23.12	26.95	0.19	0.62	0.27	0	45.49	0.61	99.74
	Esfahfo	0.51	0.13	0.26	0.5	0.52	0.46	22.93	25.73	0.14	0.6	0.14	0	47.54	0.45	99.91
	Farsfars	0.67	0.16	0.41	0.28	0.75	0.61	21.74	27.89	0.27	0.7	0.27	0.15	45.08	0.6	99.58
	Farsmar	0	0	0	0	0	0	5.35	5.64	0	0	0	0	89.01	0	100
	Hamdan	0.82	0.2	0.47	0.33	0.79	0.64	25.9	30.51	0.15	0.83	0.19	0.12	37.54	1.09	99.58
	Khorbir	0.2	0.05	0.14	0.4	0.21	0.19	13.98	17.16	0.26	0.33	0.25	0.29	64.76	1.39	99.61
	Khormo	1.11	0.65	0.24	0.87	0.82	0.51	16.48	17.52	0.18	3.93	0.2	0.1	41.42	0.6	84.63
	Qazvin	0.68	0.17	0.37	0.67	0.68	0.51	23.51	26.08	0.19	0.67	0.4	0.16	44.92	0.68	99.69
	Yazd	0.41	0.13	0.19	0.97	0.41	0.36	20.37	22.38	0.25	0.67	0.25	0.1	51.78	1.21	99.48
	Yazshah	0.57	0.14	0.33	0.39	0.63	0.54	23.43	30.15	0	0.64	0.18	0	41.24	0.67	98.91

* Ri^a^: Calculated retention index; RI^b^: Literature (NIST20) retention index.

The biplot of the essential oil compounds in moderate conditions implied that two main components explained 65.77% of the variation (Figure 1b). Component 1 included myrcene, α-thujene, α-terpinene, essential oil yield, and β-pinene with positive values and TFC with a negative value. Component 2 explained 20.62% of the variation, which included γ-terpinene and *p*-cymene with positive values and thymol, carvacrol, and TPC with negative values. The studied populations were divided into six groups. In the first group, Arak and Yazshah possessed high levels of *p*-cymene and γ-terpinene. Ardebil and Hamdan were classified in the second and third groups, respectively. In the fourth group, Farsfars and Esfahfo revealed a negative value of component 1, and the TFC was the major indicator, while in the fifth group, Arakkho and Khorbir populations showed low amounts of this component. Finally, the last group, Qazvin, Yazd, Farsmar, and Khormo, were classified as the high thymol, carvacrol, and TP group.

The biplot of oil compounds in severe conditions implied that two main components explained 65.07% of the variation (Figure 1c). Component 1 included myrcene, α-thujene, α-terpinene, β-pinene, γ-terpinene, and *p*-cymene with positive values, and thymol with a negative value. Component 2 explained 20.18% of the changes and included β-pinene, β-thujone, TFC, and TPC with positive values and essential oil yield with a negative value. The studied populations were divided into six groups. In the first group, Khormo possessed a positive value of components 1 and 2, while thymol and essential oil yield were low in this population. In second group, Khorbir had low amounts of thymol. Yazd was categorized in the third group. Finally, most of the populations were classified in the sixth group.

**Figure 1 foods-11-03084-f001:**
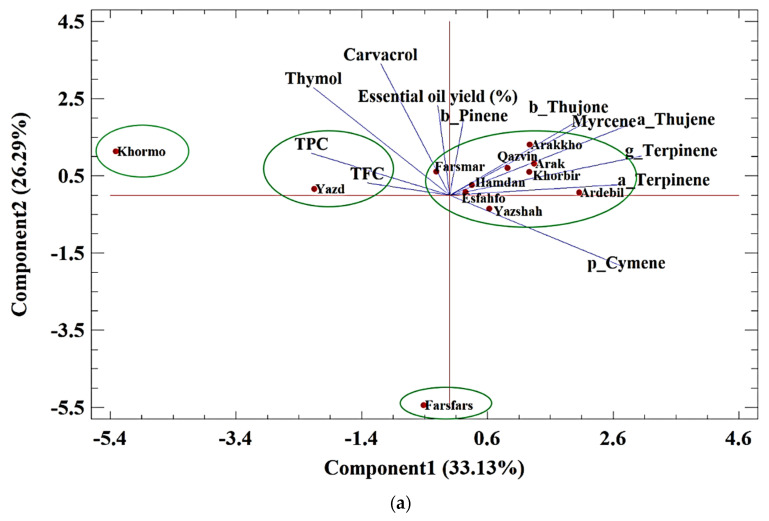
(**a**–**c**). PCA analysis for classification of 12 ajowan populations based on bioactive compounds in (**a**) normal conditions, (**b**) moderate drought stress conditions, and (**c**) severe stress conditions.

### 3.5. Antioxidant Activity

#### 3.5.1. DPPH Method

Antioxidant activity, as measured by DPPH assay and expressed in EC_50_, is shown in Figure 2. A high EC_50_ value in a sample indicates its low antioxidant activity. Water stress highly affected antioxidant capacity in ajowan but led either to increased or to decreased activity, depending on the stress level and the population of origin. The lowest EC_50_ levels (371.5 and 492.6 μg mL^−1^) were obtained under moderate and severe drought stress conditions, while these levels ranged between 506 μg mL^−1^ and 3924 μg mL^−1^ under the normal treatment (Figure 2). When subjected to the moderate drought stress treatment, 12 populations of the total studied exhibited different responses so that populations such as Yazd, Yazshah, and Farsfars showed increases in their EC_50_ levels, while Khorbir, Qazvin, Farsmar, Khormo, Arak, Arakkho, and Ardebil showed decreases in their EC_50_. Under the severe drought stress treatment, Yazd (5290 μg mL^−1^), Yazshah (1387 μg mL^−1^), and Qazvin (5080 μg mL^−1^) populations exhibited increases in EC_50_ and by the results reported in studies of Apiaceae showed elevated antioxidant capacity in response to water stress, mostly due to the suppression of the effects of reactive oxygen species (ROS) and free radicals released during drought stress [37,38]. However, the non-directional response of ajowan suggests more complex mechanisms of antioxidant regulation.

#### 3.5.2. Fe-Reducing Power

The moderate water stress treatment led to higher antioxidant activity (Figure 3a–d). The ferric thiocyanate (FTC) model revealed the highest and lowest antioxidant activities in Qazvin and Khorbir populations, respectively. In some populations, severe water stress was observed to elevate antioxidant capacity. Similar results have been reported on *Cuminum cyminum* extract from the Apiaceae family [38]. Most researchers in the field attributed the reduced antioxidant capacity to the phenolic components of seed extracts [39,40,41].

### 3.6. Total Phenolic (TPC) and Flavonoid Content (TFC)

TPC and TFC in the different ajowan populations showed high variations in response to the different levels of drought stress (Table 1). Improved TPC levels were observed in Esfahfo and Yazd populations subjected to moderate drought stress. These results were supported by those reported on TPC in *Trachyspermum ammi* that was highly influenced by the level of applied water stress.

Production of ROS in plants has been reported to be a phenomenon associated with severe drought stress [42]. It is claimed that increased TPC and TFC under drought stress might be the result of ROS accumulation in plants [37]. Polyphenolic compounds have been shown to overcome ROS effects due to their hydroxyl groups and high antioxidant activity [43].

### 3.7. HPLC Results

The HPLC analysis of phenolics extracts obtained from ajowan populations under different irrigation regimes is illustrated in Table 3. Accordingly, eight major polyphenolic compounds were recorded, including phenolic acids, viz., chlorogenic acid, gallic acid, caffeic acid, *p*-coumaric acid, ferulic acid, and rosmarinic acid, and flavonoids such as rutin and apigenin (Table 3). Clearly, high chemical polymorphisms were observed among the Iranian ajowan populations with respect to chlorogenic, caffeic, and ferulic acids contents as the major components. The amount of chlorogenic acid ranged from 3.750 ± 0.4 mg/100 g in Farsmar in normal conditions to 47.355 ± 1.6 mg/100 g in Arak in normal conditions. The highest (40.099 ± 1.6 mg/100 g) and lowest (13.225 ± 0.6 mg/100 g) amounts of caffeic acid belonged to Yazd (S) and Khormo (S), respectively. Ferulic acid content varied from 40.098 ± 1.2 mg/100 g in Hamdan (S) to 11.254 ± 0.6 mg/100 g in Farsfars (N).

#### Biplot Analysis of Polyphenolic Compounds

The biplot of the HPLC compounds in the control condition implied that two main components justified 49.85% of the variances (Figure 4a). Component 1 explained 29.68% of the changes that included gallic acid, chlorogenic acid, and *p*-coumaric with positive values, while TPC and TFC had negative values. Component 2 with the justification of 20.17% of the variances included caffeic acid, rutin, apigenin with apositive values, and ferulic acid with a negative value. The identified compounds were divided into six groups. Group one included Khorbir, Farsfars, Yazd, Esfahfo, Arakkho, and Ardebil that had a balance of all compounds. The second group included high levels of gallic acid, chlorogenic acid, and *p*-coumaric in Arak and Yazdshah. In the third group, Hamedan had high levels of ferulic acid. The next group, Qazvin, had high amounts of four compounds, including rutin, rosmarinic acid, caffeic acid, and apigenin. The fifth group, Farsmar, had high amounts of TPC and TFC. Khormo was in the sixth group and had high levels of ferulic acid, TFC, and TPC.

A biplot of medium conditions could explain about 46% of the changes. In the first component, gallic acid, *p*-coumaric acid, and ferulic acid had positive values, and TFC had a negative value (Figure 4b). The second component included chlorogenic acid, caffeic acid, TFC, and apigenin. The studied populations were divided into five groups. The first group covering Yazd, Qazvin, and Farsmar included TPC, caffeic acid, and chlorogenic acid. In the next group, Arakkho, Hamedan, Khorbir, Yazdshah, and Esfahfo, had compounds such as ferulic, rosmarinic, gallic, and *p*-coumaric acids with high percentages. The Farsfars in the third group had high TFC. Similarly, Ardebil, Arak, and Khormo were classified in the next group, and Ardabil was grouped the last group.

In severe conditions, the biplot described 48% of the changes (Figure 4c). Component 1 included gallic, *p*-coumaric, and ferulic acid, as well as TPC and TFC with positive values, and component 2 with a justification of 23.18% of compounds such as rutin, rosmarinic acid, and chlorogenic acid had a negative value. The studied populations were divided into five groups. In the first group, Khormo had high levels of gallic acid, *p*-coumaric, ferulic acid, TPC, and TFC. In the second group, Arakkho had rutin, apigenin, and rosmarinic acid. In the central group, Arak, Yazd, Esfahfo, Yazshah, Farsfars, Farsmar, and Qazvin were with negative values of component 1 and positive values of component 2. The fourth group included Hamedan and Khorbir, and the last group included Ardebil.

**Table 3 foods-11-03084-t003:** Major phenolic and flavonoid compounds of studied Ajowan populations based on HPLC analysis. The values are expressed as mg/100 g of sample dry weight. The results are expressed as mean ± SD.

Standards		Gallic Acid	Chlorogenic Acid	Caffeic Acid	*p*-Coumaric Acid	Rutin	Ferulic Acid	Rosmarinic Acid	Apigenin
RT *		4.692	12.315	13.13	26.548	28.927	29.375	36.257	56.356
Stress	Populations								
Normal	Arak	12.850 ± 0.2 ^j^	47.355 ± 1.6 ^a^	14.446 ± 0.8 ^qr^	18.722 ± 0.6 ^i^	14.949 ± 0.4 ^k^	20.481 ± 0.4 ^n^	2.996 ± 0.2 ^hij^	8.775 ± 0.2 ^ghi^
	Arakkho	8.544 ± 0.4 ^t^	8.499 ± 0.4 ^x^	13.246 ± 0.4 ^w^	16.816 ± 0.4 ^mn^	13.477 ± 0.3 ^mn^	16.525 ± 0.4 ^t^	3.721 ± 0.3 ^de^	8.412 ± 0.3 ^klm^
	Ardebil	16.103 ± 0.4 ^d^	12.623 ± 0.4 ^q^	18.877 ± 0.5 ^i^	16.047 ± 0.4 ^p^	12.724 ± 0.4 ^r^	13.434 ± 0.5 ^y^	2.534 ± 0.2 ^lm^	8.637 ± 0.3 ^ij^
	Esfahfo	13.945 ± 0.3 ^h^	8.085 ± 0.3 ^y^	15.288 ± 0.6 ^no^	16.422 ± 0.2 ^o^	13.598 ± 0.2 ^m^	17.043 ± 0.7 ^s^	2.820 ± 0.2 ^jk^	8.377 ± 0.2 ^m^
	Farsfars	8.855 ± 0.6 ^rs^	22.495 ± 0.4 ^h^	13.635 ± 0.2 ^uv^	17.958 ± 0.6 ^jk^	13.120 ± 0.6 ^op^	11.254 ± 0.6 ^z^	2.883 ± 0.1 ^ijk^	8.741 ± 0.6 ^hi^
	Farsmar	6.152 ± 0.4 ^w^	3.750 ± 0.4 ^b^	13.977 ± 0.4 ^t^	15.575 ± 0.5 ^r^	19.282 ± 0.5 ^b^	14.602 ± 0.5 ^x^	2.445 ± 0.2 ^mno^	9.033 ± 0.3 ^ef^
	Hamdan	11.694 ± 0.5 ^l^	7.127 ± 0.5 ^z^	13.569 ± 0.4 ^v^	15.990 ± 0.5 ^pq^	11.783 ± 0.8 ^w^	32.799 ± 0.7 ^c^	2.224 ± 0.3 ^pqrs^	8.477 ± 0.4 ^jklm^
	Khorbir	9.876 ± 0.4 ^p^	9.445 ± 0.3 ^w^	14.060 ± 0.6 ^st^	18.197 ± 0.6 ^j^	16.669 ± 0.6 ^g^	21.156 ± 0.7 ^m^	5.134 ± 0.2 ^b^	8.626 ± 0.3 ^ijk^
	Khormo	9.398 ± 0.5 ^q^	11.634 ± 0.5 ^s^	18.419 ± 0.4 ^j^	14.605 ± 0.5 ^t^	12.229 ± 0.6 ^t^	22.381 ± 0.5 ^k^	2.291 ± 0.2 ^nopqr^	8.763 ± 0.3 ^ghi^
	Qazvin	14.380 ± 0.3 ^g^	10.791 ± 0.4 ^u^	23.947 ± 0.8 ^d^	17.269 ± 0.7 ^l^	19.424 ± 0.4 ^b^	15.769 ± 0.5 ^u^	3.313 ± 0.3 ^f^	9.184 ± 0.3 ^de^
	Yazd	11.657 ± 0.4 ^l^	6.984 ± 0.4 ^z^	21.173 ± 0.7 ^f^	17.400 ± 0.3 ^l^	13.306 ± 0.3 ^no^	18.333 ± 0.4 ^qr^	2.196 ± 0.1 ^qrs^	8.439 ± 0.4 ^ijklm^
	Yazshah	15.417 ± 0.7 ^e^	36.003 ± 1.4 ^d^	14.285 ± 0.4 ^rs^	21.725 ± 0.3 ^c^	15.939 ± 0.6 ^i^	26.630 ± 0.9 ^g^	2.291 ± 0.2 ^nopqr^	8.383 ± 0.5 ^lm^
Medium	Arak	8.969 ± 0.3 ^r^	15.718 ± 0.7 ^m^	17.183 ± 0.6 ^l^	14.736 ± 0.7 ^t^	14.897 ± 0.6 ^k^	15.044 ± 0.5 ^w^	3.175 ± 0.2 ^fgh^	9.289 ± 0.5 ^cd^
	Arakkho	11.773 ± 0.7 ^l^	16.842 ± 0.5 ^kl^	19.545 ± 0.4 ^h^	17.811 ± 0.8 ^k^	12.478 ± 0.3 ^s^	22.754 ± 0.7 ^j^	4.539 ± 0.4 ^c^	8.958 ± 0.5 ^fgh^
	Ardebil	10.366 ± 0.5 ^o^	11.617 ± 0.5 ^s^	22.992 ± 0.7 ^m^	16.710 ± 0.6 ^n^	15.343 ± 0.5 ^j^	15.782 ± 0.5 ^u^	3.242 ± 0.3 ^fg^	8.424 ± 0.3 ^jklm^
	Esfahfo	12.927 ± 0.4 ^j^	11.102 ± 0.7 ^t^	18.260 ± 0.3 ^j^	16.975 ± 0.6 ^m^	15.469 ± 0.6 ^j^	25.144 ± 0.7 ^h^	3.907 ± 0.3 ^d^	8.487 ± 0.3 ^jklm^
	Farsfars	13.936 ± 0.5 ^h^	11.681 ± 0.4 ^s^	21.276 ± 0.6 ^f^	15.775 ± 0.5 ^qr^	18.029 ± 0.6 ^d^	16.884 ± 0.5 ^s^	3.094 ± 0.2 ^ghi^	9.927 ± 0.4 ^a^
	Farsmar	13.284 ± 0.6 ^i^	37.070 ± 1.4 ^c^	14.622 ± 0.3 ^pq^	19.167 ± 0.6 ^h^	12.149 ± 0.4 ^tu^	18.164 ± 0.5 ^r^	2.838 ± 0.1 ^jk^	9.566 ± 0.5 ^b^
	Hamdan	14.122 ± 0.3 ^h^	14.883 ± 0.6 ^n^	15.280 ± 0.6 ^no^	17.734 ± 0.7 ^k^	12.505 ± 0.3 ^s^	22.477 ± 0.7 ^k^	2.430 ± 0.2 ^mnop^	8.599 ± 0.5 ^ijkl^
	Khorbir	12.364 ± 0.6 ^k^	13.275 ± 0.5 ^p^	14.720 ± 0.7 ^p^	19.059 ± 0.7 ^h^	11.931 ± 0.3 ^uvw^	22.548 ± 0.7 ^jk^	5.658 ± 0.2 ^a^	8.461 ± 0.4 ^jklm^
	Khormo	5.366 ± 0.5 ^x^	6.488 ± 0.4 ^a^	15.139 ± 0.7 ^no^	15.179 ± 0.5 ^s^	12.831 ± 0.4 ^qr^	15.299 ± 0.4 ^v^	2.477 ± 0.2 ^mn^	8.968 ± 0.4 ^efg^
	Qazvin	14.000 ± 0.6 ^h^	16.705 ± 0.6 ^l^	30.524 ± 1.4 ^b^	17.967 ± 0.6 ^jk^	13.901 ± 0.4 ^l^	18.444 ± 0.5 ^q^	5.330 ± 0.3 ^b^	9.439 ± 0.3 ^bc^
	Yazd	10.629 ± 0.7 ^n^	12.205 ± 0.4 ^r^	27.637 ± 0.6 ^c^	16.822 ± 0.7 ^mn^	14.948 ± 0.6 ^k^	21.178 ± 0.7 ^m^	3.289 ± 0.2 ^fg^	8.418 ± 0.3 ^jklm^
	Yazshah	19.172 ± 0.5 ^b^	9.532 ± 0.4 ^vw^	18.210 ± 0.4 ^j^	21.332 ± 0.7 ^d^	16.264 ± 0.4 ^h^	23.578 ± 0.9 ^i^	2.383 ± 0.2 ^mnopq^	8.511 ± 0.5 ^jklm^
Severe	Arak	16.256 ± 0.5 ^d^	24.053 ± 1.3 ^g^	16.829 ± 0.7 ^m^	19.573 ± 0.7 ^g^	11.859 ± 0.6 ^vw^	21.430 ± 0.9 ^l^	2.136 ± 0.2 ^rst^	8.610 ± 0.4 ^ijk^
	Arakkho	14.068 ± 0.7 ^h^	11.486 ± 0.6 ^sh^	13.869 ± 0.3 ^tu^	27.151 ± 1.5 ^a^	19.986 ± 0.8 ^a^	38.352 ± 1.7 ^b^	3.591 ± 0.3 ^e^	9.524 ± 0.5 ^b^
	Ardebil	8.658 ± 0.3 ^st^	9.698 ± 0.8 ^sh^	17.578 ± 0.5 ^k^	16.167 ± 0.5 ^op^	18.397 ± 0.3 ^c^	14.807 ± 0.3 ^x^	3.235 ± 0.2 ^fg^	8.628 ± 0.3 ^ijk^
	Esfahfo	11.681 ± 0.6 ^l^	20.288 ± 1.2 ^i^	16.800 ± 0.5 ^m^	20.664 ± 0.5 ^e^	13.024 ± 0.4 ^pq^	30.924 ± 1.4 ^d^	2.529 ± 0.3 ^lm^	8.607 ± 0.3 ^ijk^
	Farsfars	8.694 ± 0.4 ^st^	35.553 ± 2.3 ^e^	13.974 ± 0.5 ^t^	17.786 ± 0.6 ^k^	15.434 ± 0.5 ^j^	26.638 ± 0.5 ^g^	2.739 ± 0.2 ^kl^	9.062 ± 0.4 ^ef^
	Farsmar	7.657 ± 0.4 ^v^	13.902 ± 0.6 ^o^	14.024 ± 0.4 ^st^	19.506 ± 0.9 ^g^	12.062 ± 0.2 ^tuv^	19.469 ± 0.4 ^o^	2.243 ± 0.3 ^opqrs^	8.463 ± 0.3 ^jklm^
	Hamdan	17.354 ± 0.5 ^c^	13.950 ± 0.8 ^o^	15.039 ± 0.8 ^o^	19.143 ± 0.7 ^h^	17.772 ± 0.7 ^e^	40.098 ± 1.2 ^a^	1.939 ± 0.2 ^tu^	8.476 ± 0.5 ^jklm^
	Khorbir	11.316 ± 0.6 ^m^	15.065 ± 0.9 ^n^	13.683 ± 0.6 ^uv^	21.146 ± 0.8 ^d^	12.202 ± 0.7 ^t^	18.894 ± 0.3 ^p^	3.351 ± 0.3 ^f^	8.557 ± 0.5 ^ijklm^
	Khormo	37.695 ± 2.4 ^a^	16.944 ± 0.8 ^k^	13.225 ± 0.6 ^w^	23.483 ± 0.9 ^b^	11.741 ± 0.7 ^w^	29.615 ± 0.6 ^e^	2.099 ± 0.2 ^u^	8.409 ± 0.4 ^klm^
	Qazvin	8.291 ± 0.7 ^u^	27.190 ± 1.4 ^f^	20.361 ± 1.2 ^g^	17.880 ± 0.6 ^k^	18.128 ± 0.7 ^d^	20.474 ± 0.4 ^n^	2.209 ± 0.3 ^qrs^	9.506 ± 0.1 ^bc^
	Yazd	14.615 ± 0.6 ^f^	43.357 ± 2.3 ^b^	40.099 ± 1.6 ^a^	20.042 ± 0.9 ^f^	17.316 ± 0.6 ^f^	28.522 ± 0.4 ^f^	2.041 ± 0.4 ^st^	8.582 ± 0.5 ^ijklm^
	Yazshah	8.183 ± 0.4 ^u^	20.017 ± 1.3 ^j^	15.344 ± 0.3 ^n^	19.991 ± 0.9 ^f^	20.123 ± 0.8 ^a^	26.509 ± 0.5 ^g^	2.165 ± 0.1 ^rs^	8.585 ± 0.4 ^ijklm^

* The data were sorted based on the retention time (RT) of components. The letters are based on least significant difference (LSD) test in 5% probability level.

**Figure 4 foods-11-03084-f004:**
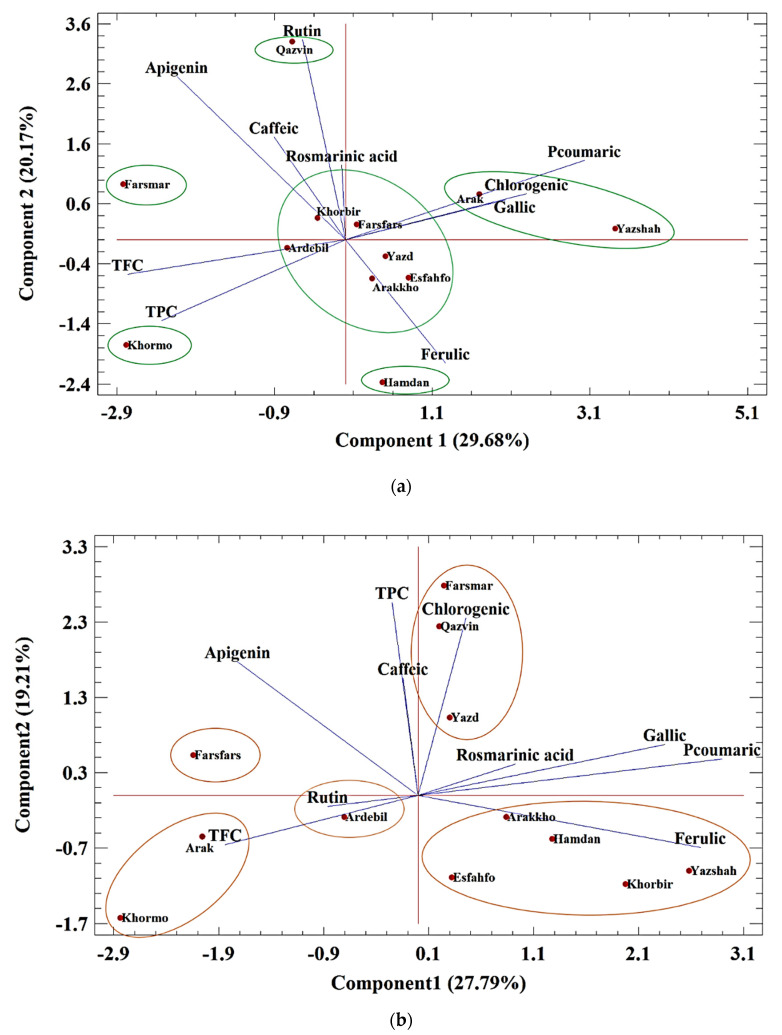
(**a**–**c**). PCA analysis for classification of 12 ajowan populations based on HPLC compounds in (**a**) normal, (**b**) moderate drought stress, and (**c**) severe stress conditions.

### 3.8. Cluster Analysis of All Compounds

For better interpretation and conclusions of all data including oil content, HPLC, and GC–MS results, cluster analysis was performed in three water deficit levels (Figure 5). Accordingly, two main clusters were determined. Cluster 1 included normal conditions. Cluster 1 classified the populations into two subgroups. Subgroup 1 included the populations with high amounts of essential oil yield and chlorogenic acid, while subgroup 2 consisted of ajowan with high amounts of TPC and TFC. The second main cluster also included two subgroups including the samples in moderate and in moderate + severe (M + S) conditions. The moderate subgroup mainly had high rosmarinic acid content, while rutin, γ-terpinene, caffeic acid, *p*-coumaric, and α-terpinene compounds were the major components in the M + S subgroup.

## 4. Discussion

The biosynthesis of secondary metabolites is regulated by genetic mechanisms as well as environmental factors. One hypothesis often put forth to explain the effects of environmental factors on plant secondary metabolites is the lack of balance between photosynthesis and growth in plants [44]. It is claimed, for instance, that moderate drought, moderate nutrient limitation, and low temperature might lead to an increased carbon pool available for the synthesis of secondary metabolites. Moreover, variations in essential oil have also been interpreted with recourse to temperature [45], as evidenced by the reported structural degradation of secretory hairs and glands and the consequent collapse of epithelial cells in the samples dried due to exposure to high temperatures [46]. It might be similarly claimed that oil content in ajowan seeds declines as a result of high temperatures and drought.

**Figure 5 foods-11-03084-f005:**
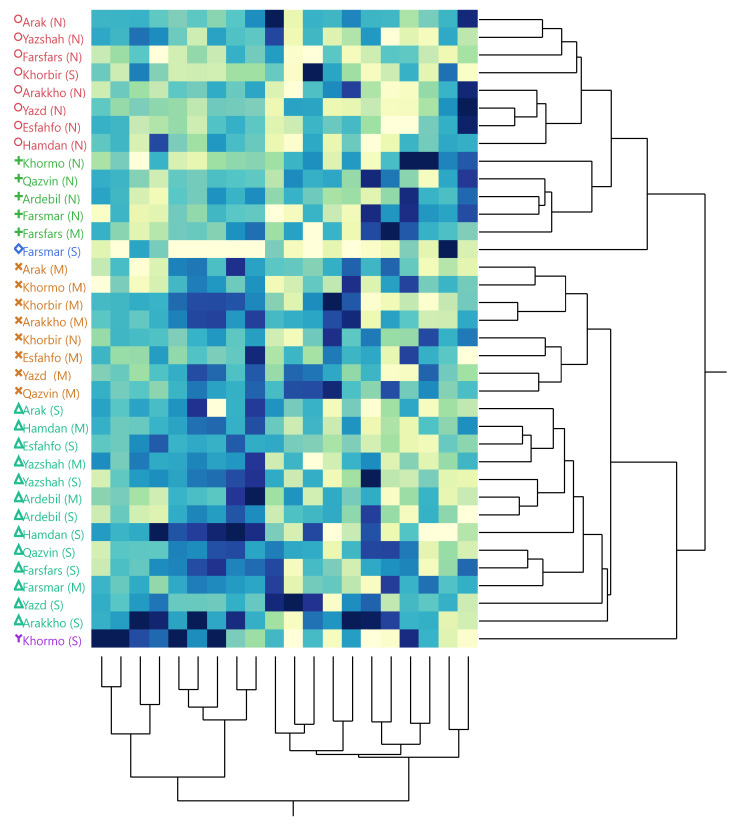
Hierarchical clustering analysis (HCA) of 12 ajowan populations based on oil content, HPLC, and GC–MS compounds in three water deficit levels.

Thus, chemical composition might be affected both by genetic and environmental factors. Both mechanisms are involved in the accumulation of secondary metabolites in plants because of drought stress [44]. When plants are subjected to high temperatures along with drought, reactive oxygen species (ROS), which are toxic components that lead to oxidative destruction of cells, might accumulate in higher amounts, giving rise to oxidative stress [47]. In this situation, elevated levels of ROS under stress cause lipid peroxidation [48], which is a natural process in plants activated in response to stress.

The other mechanism is the one proposed in [49], when drought causes stomata closure, which in turn decreases CO_2_ uptake. The changes in CO_2_ fixation cause a remarkable decrease in the use of reductive components such as NADPH^+^H^+^, leading to the production of high amounts of NADPH^+^H^+^ and superoxide radicals. According to the law of mass action, any increase in reductive power causes changes in all the reactions involving NADPH^+^H^+^ consumption on the reduction side of the equilibrium. Thus, the biosynthesis of highly reduced secondary components increases in a highly reduced medium, and hence the increased biosynthesis of highly reduced precursors such as isoprenoids.

The mechanisms just described can be most observed in the leaves of such plants as thyme [34]. In the present study, a decrease was observed in the monoterpene content in ajowan seeds. Monoterpenes are metabolites most frequently and commonly detected in the essential oils of plants, as is thymol in ajowan oil. Moreover, their quantities highly depend on drought conditions and vary with plant organ. Increased emission of monoterpenes under drought conditions in some plant species might lead to confused storage of sesquiterpenes in the leaves [50]. In the present study, for instance, the amount of thymol decreased as a result of drought stress. This can be explained by the fact that transpiration leads to increased monoterpene emissions, with elevated levels expected to be observed at the beginning of the drought period. In ajowan, drought duration and temperature were much higher during the seed harvesting period than during the vegetative phases. Thymol, an aromatic monoterpene, occurs in higher amounts during the long drought periods (of seed harvesting) than during the short drought periods (of leaf harvesting). Accordingly, it might be hypothesized that accumulation of a constitutive defence within plants usually conforms to expectations of the ‘optimal defence theory’ (ODT), which suggests that the highest protection level can be observed in the organs with the highest fitness value. These organs might vary with plant species or the type of metabolites, as reported for terpenes [51], alkaloids [52,53], and phenylpropanoids [54,55].

In the present research, for the first time, HPLC analysis of ajowan plants from several populations was performed in response to water stress. In normal conditions, high variation was observed in polyphenolic compounds. Phenolic acids were the most abundant compounds. There is only one report regarding the phenolic acids in ajowan in response to laser treatments [56]. These researchers also revealed high variation in polyphenolic compounds. However, they reported lower amounts for most phenolic compounds. In the present study, apigenin and rutin were the major flavonoids in the ajowan seeds. These flavonoids were also reported in [56]. In a previous report, chlorogenic acid was the most predominant constituent [57]. In agreement with the present research data, rutin was the most abundant flavonoid in their study, while gallic acid, caffeic acid, and ferulic acid were not detected. The phenolic acids of two local varieties of ajowan were reported in [58]. In their research, gallic acid and *p*-coumaric acid were the major polyphenolic components. The changes in phenolic compounds and oil components of ajowan under different fertilizer treatments were reported in [59]. Accordingly, chlorogenic, coumaric, and rosmarinic acids were the major compounds, which is in line with the results of the present study.

In most cases, water stress elevated phenolic acids and decreased the flavonoids in severe stress condition (Table 1). Similar trends were also obtained for *Achillea pachycephala* [29], *Cynara cardunculus* [60], and *Ctenanthe setosa* [61]. Chlorogenic acid showed different trends in 12 studied populations, which also revealed such variations in different tarragon populations subjected to water stress [62]. The increase in phenolic compounds as a result of drought stress can be interpreted as an increase in cell wall lignification [60].

In most of the previous research, drought stress led to decreases in flavonoid contents in the plants [60,63]. Similar trends were also obtained for rutin and apigenin in this study. Ajowan seeds are edible, and flavonoid content should be monitored along with thymol content of the oil as crucial nutraceutical components. Thus, a moderate water stress condition was the best environment for improving these components. Moreover, the antioxidant capacity of polyphenolic compounds highly depends on the presence of hydroxyl groups (Figure 2). The presence of one hydroxyl group in the B ring of apigenin might be the reason for its lower antioxidant potential against ROS in severe drought condition [63]. A high variation was observed regarding antioxidant capacity in different studied populations as compared with a synthetic antioxidant (BHT). This might have originated from both genetic and epigenetic factors. Moreover, the response of populations to drought conditions was different in terms of antioxidant capacity. This might be due the fact that antioxidant enzymes mostly are responsible for overcoming the accumulation of ROS in drought conditions, and antioxidant capacity mostly represents total evaluated antioxidant values.

## 5. Conclusions

In this research, the effects of different drought conditions were assessed on 12 ajowan populations. The present research highlighted the accumulation pattern of phytochemicals in the seeds of ajowan as a valuable industrial medicinal plant. The studied populations as well as their metabolites revealed different responses to water stress. Thymol content was mostly elevated with severe stress, while oil content showed variable trends in different populations. Among them, the Yazd population can be introduced as an elite stock with low susceptibility to thymol and oil content decrease. Moreover, Arak and Khormo populations showed the highest accumulation of polyphenolic compounds. Finally, the moderate water stress condition can be suggested for improving the health benefit components such as thymol and flavonoids in the seeds of ajowan. Consequently, the results of the present research can provide insights for use of selected populations for further mass cultivation of this valuable medicinal plant.

## Figures and Tables

**Figure 2 foods-11-03084-f002:**
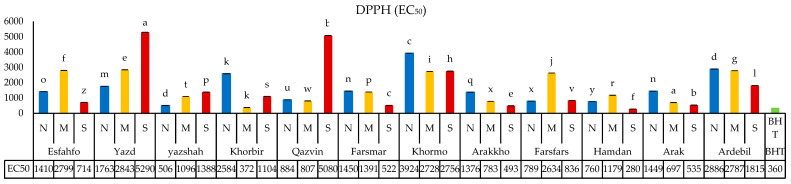
EC_50_ of 12 ajowan population extracts based on DPPH assay in comparison to BHT. (N) normal, (M) moderate, and (S) severe drought conditions.

**Figure 3 foods-11-03084-f003:**
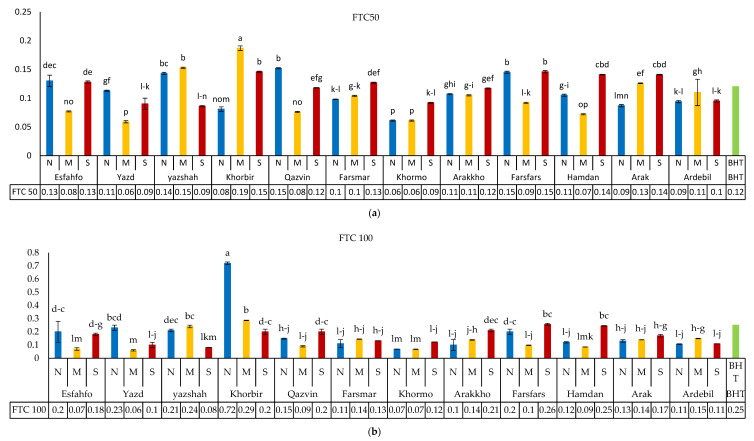
(**a**) Antioxidant capacity based on reducing power in 50 ppm conc. of 12 ajowan extracts as compared to BHT. (N) Normal, (M) moderate, and (S) severe drought conditions. (**b**) Antioxidant capacity based on reducing power in 100 ppm conc. of 12 ajowan extracts as compared to BHT. (N) Normal, (M) moderate, and (S) severe drought conditions. (**c**) Antioxidant capacity based on reducing power in 300 ppm conc. of 12 ajowan extracts as compared to BHT. (N) Normal, (M) moderate, and (S) severe drought conditions. (**d**) Antioxidant capacity based on reducing power in 500 ppm conc. of 12 ajowan extracts as compared to BHT. N) Normal, (M) moderate, and (S) severe drought conditions.

**Table 1 foods-11-03084-t001:** Contents of total phenolics, total flavonoids, and essential oil yield of ajowan populations. The values are given as mean ± SD.

Stress	Populations	TFC* (mg QE g^−1^ DW)	TPC** (mg TAE g^−1^ DW)	Essential Oil Yield (%)
Normal	Arak	2.8 ± 0.06 ^mn^	33.15 ± 0.32 ^hi^	5.15 ± 0.06 ^a^
	Arakkho	1.2 ± 0.05 ^pq^	28.65 ± 0.41 ^ij^	5.22 ± 0.06 ^a^
	Ardebil	9.7 ± 0.63 ^c^	45.07 ± 3.46 ^f^	3.85 ± 0.05 ^e^
	Esfahfo	0.7 ± 0.05 ^q^	33.32 ± 0.54 ^hi^	5.37 ± 0.1 ^a^
	Farsfars	4.00 ± 0.08 ^ghi^	48.09 ± 0.99 ^jkl^	2.1 ± 0.15 ^mno^
	Farsmar	9.5 ± 0.49 ^c^	68.69 ± 0.72 ^e^	4.33 ± 0.10 ^c^
	Hamdan	4.6 ± 0.24 ^hij^	33.84 ± 0.53 ^hi^	2.79 ± 0.08 ^ghi^
	Khorbir	3.1 ± 0.15 ^mn^	119.30 ± 10.11 ^b^	3.61 ± 0.15 ^e^
	Khormo	10.6 ± 0.18 ^a^	171.36 ± 2.13 ^a^	4.08 ± 0.03 ^cd^
	Qazvin	3.3 ± 0.24 ^mn^	12.04 ± 0.87 ^op^	4.73 ± 0.15 ^b^
	Yazd	1.3 ± 0.14 ^pq^	25.84 ± 0.57 ^jk^	5.55 ± 0.2 ^a^
	Yazshah	1.9 ± 0.08 ^op^	12.70 ± 0.63 ^op^	1.68 ± 0.16 ^lmn^
Medium	Arak	3.9 ± 0.08 ^ijk^	13.14 ± 0.34 ^nop^	1.27 ± 0.01 ^p^
	Arakkho	4.1 ± 0.05 ^hij^	21.33 ± 0.12 ^klm^	2.9 ± 0.11 ^g^
	Ardebil	3 ± 0.18 ^lmn^	45.07 ± 3.46 ^lm^	3.85 ± 0.05 ^mno^
	Esfahfo	8.5 ± 0.23 ^c^	48.71 ± 0.21 ^f^	0.7 ± 0.03 ^r^
	Farsfars	7.6 ± 0.07 ^d^	48.09 ± 0.99 ^f^	2.1 ± 0.15 ^jk^
	Farsmar	4.7 ± 0.08 ^fg^	91.66 ± 0.93 ^c^	2.8 ± 0.05 ^gh^
	Hamdan	1.7 ± 0.08 ^p^	38.71 ± 0.60 ^g^	3.34 ± 0.06 ^f^
	Khorbir	2.6 ± 0.29 ^no^	9.29 ± 0.13 ^p^	2.22 ± 0.21 ^ij^
	Khormo	8.4 ± 0.43 ^cd^	33.61 ± 0.81 ^hi^	2.21 ± 0.16 ^jk^
	Qazvin	3.7 ± 0.11 ^ijkl^	76.38 ± 0.69 ^d^	2.51 ± 0.14 ^hij^
	Yazd	1.2 ± 0.05 ^pq^	100.55 ± 0.61 ^b^	1.8 ± 0.03 ^lm^
	Yazshah	3.4 ± 0.05 ^klm^	17.69 ± 0.03 ^mn^	2.6 ± 0.05 ^ghi^
Severe	Arak	5.2 ± 0.14 ^f^	11.97 ± 0.41 ^nop^	1.67 ± 0.04 ^mno^
	Arakkho	4.7 ± 0.14 ^fgh^	49.21 ± 1.18 ^f^	1.5 ± 0.06 ^mnop^
	Ardebil	1.3 ± 0.05 ^p^	47.32 ± 1.24 ^f^	0.7 ± 0.05 ^r^
	Esfahfo	1.7 ± 0.03 ^p^	13.56 ± 0.65 ^nop^	2 ± 0.05 ^kl^
	Farsfars	6.3 ± 0.11 ^e^	12.90 ± 0.64 ^nop^	3.87 ± 0.12 ^de^
	Farsmar	3.5 ± 0.20 ^jklm^	17.39 ± 1.40 ^mn^	1.4 ± 0.14 ^op^
	Hamdan	4.2 ± 0.26 ^ghi^	7.00 ± 1.67 ^p^	1.72 ± 0.014 ^lmn^
	Khorbir	4.8 ± 0.17 ^fgh^	17.36 ± 0.22 ^mno^	0.91 ± 0.04 ^qr^
	Khormo	9.6 ± 0.14 ^b^	45.54 ± 2.57 ^f^	0.84 ± 0.03 ^r^
	Qazvin	6.6 ± 0.63 ^e^	16.95 ± 2.88 ^mno^	1.44 ± 0.09 ^nop^
	Yazd	4.4 ± 0.43 ^ghi^	36.87 ± 0.57 ^gh^	1.57 ± 0.22 ^mno^
	Yazshah	2.4 ± 0.02 ^no^	32.95 ± 0.61 ^hi^	1.21 ± 0.06 ^pq^

* Total flavonoids content; ** total phenolics content. The letters are based on least significant difference (LSD) test in 5% probability level.

## Data Availability

Data is contained within the article or Appendix A and available upon reasonable request from the corresponding author.

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
