# Peer review of "Changes in Essential Oil Composition, Polyphenolic Compounds and Antioxidant Capacity of Ajowan (Trachyspermum ammi L.) Populations in Response to Water Deficit"

_foods, 2022, doi:10.3390/foods11193084_

Round 1
Reviewer 1 Report
Title: Changes in Essential Oil Composition, Polyphenolic Compounds and Antioxidant Capacity of Ajowan (Trachyspermum ammi L.) Populations in Response to Water Deficit
Comments to authors:
In this paper, the effects of three irrigation methods on seed yield, essential oil constituents, polyphenolic composition, and antioxidant capacity of Ajowan (Trachyspermum ammi L.) were investigated. This study is innovative and has a lot of work, but the expression of the manuscript needs to be improved. The followings are some comments and suggestions for authors to consider and improve the manuscript.
1. The first three paragraphs in the discussion section are all about mechanisms, and we suggest that the author trim this section or put it in the introduction.
2. The article has little discussion on antioxidant capacity, the author should add relevant detail discussion.
3. The conclusion section suggests adding some presentation of experimental methods and data.
4. “3.6.3. Total Phenolic (TPC) and flavonoid content (TFC)” is included in “3.6. Antioxidant activity”, which is not quite the right division.
5. Pages after page 14 are not marked with line numbers.
6. The significant analysis of total flavonoids content; total phenolics content; essential oil (Table 1), major phenolic and flavonoid compounds of studied Ajowan populations (Table 3) should be performed. The caption of Table 3 should be located in the above of Table 3.
7. The detail identification method of Volatile compounds should be added in the method section.
Author Response
We appreciate the reviewers due to scrutiny during checking the manuscript. Improvements has been done in a manuscript. The changes have been marked on blue colour of text.
In this paper, the effects of three irrigation methods on seed yield, essential oil constituents, polyphenolic composition, and antioxidant capacity of Ajowan (Trachyspermum ammi L.) were investigated. This study is innovative and has a lot of work, but the expression of the manuscript needs to be improved. The followings are some comments and suggestions for authors to consider and improve the manuscript.
The first three paragraphs in the discussion section are all about mechanisms, and we suggest that the author trim this section or put it in the introduction.
Thank you for underlining that point. We shorted slightly mentioned part.
The article has little discussion on antioxidant capacity, the author should add relevant detail discussion.
Required discussion was added into discussion part.
The conclusion section suggests adding some presentation of experimental methods and data.
We added necessary information into conclusion section.
“3.6.3. Total Phenolic (TPC) and flavonoid content (TFC)” is included in “3.6. Antioxidant activity”, which is not quite the right division.
The order was changed according to suggestion.
Pages after page 14 are not marked with line numbers.
fixed
The significant analysis of total flavonoids content; total phenolics content; essential oil (Table 1), major phenolic and flavonoid compounds of studied Ajowan populations (Table 3) should be performed. The caption of Table 3 should be located in the above of Table 3.
Changes were included in a manuscript and tables.
The detail identification method of Volatile compounds should be added in the method section.
Added

Reviewer 2 Report
The subject addressed is worthy of investigation, and some interesting results have been got. At the same time, the long-term work is worth affirming. However, a number of points need clarifying and certain statements require further justification. The detailed comments follow below:
Line 86-88, Please specify the end date and whether there are different maturity periods for different irrigation conditions.
Line 127, What is the basis for setting the concentration? Why not choose the equal scale setting?
Line 143-145, Please present standard curve results for all measured substances.
Line 240, Inconsistencies in the presentation of letters, and please check all pictures and tables.
Line 240, FTC is the first acronym that needs to be defined, either in the Methods section.
Line 285, Please pay attention to the layout problem of the picture. Some of the picture covers cannot be viewed. At the same time, the authors do not seem to present 50 ppm results for either EC50 or FTC50.
Line 286, Inconsistent abbreviations. At the same time, I suggest that TPF and TFC content be placed before antioxidant assays, as presented in the Methods section, and antioxidant activity is a confirmation of content change.
Line 313, Subsequent content has no line number.
Author Response
We appreciate the reviewers due to scrutiny during checking the manuscript. Improvements has been done in a manuscript. The changes have been marked on blue colour of text.
The subject addressed is worthy of investigation, and some interesting results have been got. At the same time, the long-term work is worth affirming. However, a number of points need clarifying and certain statements require further justification. The detailed comments follow below:
Line 86-88, Please specify the end date and whether there are different maturity periods for different irrigation conditions.
The information was added to 2.2. section.
As our purpose was to check the effect of drought stress on flowering stage, we begun the treatments at the beginning of flowering to observe the effects of stress on this stage as it was more important for secondary metabolites. We performed all the drought condition at the same time but with different water deficit condition in a RCBD experimental design.
Line 127, What is the basis for setting the concentration? Why not choose the equal scale setting?
The concentrations were chosen on the basis previous experiments and our experience, Food Chemistry, Tohidi et al. 2017, 220, 153-161
Line 143-145, Please present standard curve results for all measured substances.
Calibration curves are added to supplementary data (Fig. S1)
Line 240, Inconsistencies in the presentation of letters, and please check all pictures and tables.
It was re-checked and improved.
Line 240, FTC is the first acronym that needs to be defined, either in the Methods section.
Explained
Line 285, Please pay attention to the layout problem of the picture. Some of the picture covers cannot be viewed. At the same time, the authors do not seem to present 50 ppm results for either EC50 or FTC50.
Thank you for careful checking the manuscript. The value of 50 ppm was added. It was not visible because of overlapping of figures in the previous submission.
Line 286, Inconsistent abbreviations. At the same time, I suggest that TPF and TFC content be placed before antioxidant assays, as presented in the Methods section, and antioxidant activity is a confirmation of content change.
Done
Line 313, Subsequent content has no line number.
Done

Reviewer 3 Report
Work not too original but interesting and conducted with good scientific method.
In paragraph 2.4, GC-MS analysis,
the authors must report the injection volume, the solvent used for the dilution and the dilution factor if a dilution of the EO is carried out.
In a paragraph 2.4.1 Identification of essential oil constituents ,
the authors must specify also how they calculated the % composition of EOs reported in table 2 and add quantitation in the name of paragraph.
In paragraph 2.8 HPLC analysis
Authors should add to the description of the method the injection volume, the column model used and its dimensions, the eluents used, flowrate and the gradient used.
Figure 2 is difficult to read and understand. Authors should at least add more information to the caption or make it more readable.
In paragraph 3.6.2. Fe-reducing power,
there is an abbreviation "FTC" which is not explained anywhere in the manuscript.
In paragraph 3.7 HPLC results and table 3 the authors must report the correct format of the numbers in accordance with the value of the standard deviation.
The authors must reduce or group the number of biplots in figures 1 (a-c) and 3 (a-c) andimprove or reduce the labels in the graphs for a better visualization of the data, so that the effect of drought on the Ajowan population studied as a function of the measured parameters are evident.
Author Response
We appreciate the reviewers due to scrutiny during checking the manuscript. Improvements has been done in a manuscript. The changes have been marked on blue colour of text.
In paragraph 2.4, GC-MS analysis,
the authors must report the injection volume, the solvent used for the dilution and the dilution factor if a dilution of the EO is carried out.
Necessary information was added into 2.4.1. section.
In a paragraph 2.4.1 Identification of essential oil constituents ,
the authors must specify also how they calculated the % composition of EOs reported in table 2 and add quantitation in the name of paragraph.
Added into 2.4.1. section.
In paragraph 2.8 HPLC analysis
Authors should add to the description of the method the injection volume, the column model used and its dimensions, the eluents used, flowrate and the gradient used.
Added to materials and method section.
Figure 2 is difficult to read and understand. Authors should at least add more information to the caption or make it more readable.
Changes done.
In paragraph 3.6.2. Fe-reducing power,
there is an abbreviation "FTC" which is not explained anywhere in the manuscript.
Abbreviations explained.
In paragraph 3.7 HPLC results and table 3 the authors must report the correct format of the numbers in accordance with the value of the standard deviation.
It was done.
The authors must reduce or group the number of biplots in figures 1 (a-c) and 3 (a-c) and improve or reduce the labels in the graphs for a better visualization of the data, so that the effect of drought on the Ajowan population studied as a function of the measured parameters are evident.
As the results of PCA is the output of the software, any changes can affect the scientific values of data and this form is relevant in all papers using this analysis.

Round 2
Reviewer 1 Report
The manuscript was well revised and could be accepted for publication.